

Geoscientific
Instrumentation
Methods and
Data Systems

# Artifacts from manganese reduction in rock samples prepared by focused ion beam (FIB) slicing for X-ray microspectroscopy

**Dorothea S. Macholdt**[1,2], **Jan-David Förster**[1,2], **Maren Müller**[3], **Bettina Weber**[1,2], **Michael Kappl**[3],
**A. L. David Kilcoyne**[4], **Markus Weigand**[5], **Jan Leitner**[1,2], **Klaus Peter Jochum**[1,2], **Christopher Pöhlker**[1,2], and
**Meinrat O. Andreae**[1,2,6]

[1]Biogeochemistry, TS1 Max Planck Institute for Chemistry, 55128 Mainz, Germany
[2]Multiphase Chemistry Climate Geochemistry and Particle Chemistry, Max Planck Institute for Chemistry,
55128 Mainz, Germany
[3]Physics of Interfaces Department, Max Planck Institute for Polymer Research, Mainz, Germany
[4]Lawrence Berkeley National Laboratory, Berkeley, CA, USA
[5]Modern Magnetic Systems Department, Max Planck Institute for Intelligent Systems, Stuttgart, Germany
[6]Geology and Geophysics Department, King Saud University, Riyadh, Saudi Arabia

**Correspondence:** Jan-David Förster (jd.foerster@mpic.de) and Christopher Pöhlker (c.pohlker@mpic.de)

**Abstract.** The spatial distribution of transition metal valence states is of broad interest in the microanalysis of geological and environmental samples. An example is rock varnish, a natural manganese (Mn)-rich rock coating, whose genesis mechanism remains a subject of scientific debate. We conducted scanning transmission X-ray microscopy with near-edge X-ray absorption fine-structure spectroscopy (STXM-NEXAFS) measurements of the abundance and spatial distribution of different Mn oxidation states within the nano- to micrometer thick varnish crusts. Such microanalytical measurements of thin and hard rock crusts require sample preparation with minimal contamination risk. Focused ion beam (FIB) slicing was used to obtain $\sim$ 100–1000 nm thin wedge-shaped slices of the samples for STXM, using standard parameters. However, while this preparation is suitable for investigating element distributions and structures in rock samples, we observed artifactual modifications of the Mn oxidation states at the surfaces of the FIB slices. Our results suggest that the preparation causes a reduction of $Mn^{4+}$ to $Mn^{2+}$. We draw attention to this issue, since FIB slicing, scanning electron microscopy (SEM) imaging, and other preparation and visualization techniques operating in the kilo-electron-volt range are well-established in geosciences, but researchers are often unaware of the potential for the reduction of Mn and possibly other elements in the samples.

## 1 Introduction

Rock varnish, a thin natural crust on rock surfaces with thicknesses of up to TS2 $\sim$ 250 µm, typically consists of 5 wt %– 20 wt % Mn oxyhydroxide minerals, which cement mineral dust grains forming a hard, black coating. The relevant Mn oxidation states in varnish and other natural Mn minerals are $Mn^{2+}$, $Mn^{3+}$, and $Mn^{4+}$. Even though a large number of publications on rock varnish is available, the process of Mn oxidation and precipitation of the matrix is still under controversial debate (e.g., Liu and Dorn, 1996; DiGregorio, 2002; Perry and Kolb, 2004; Thiagarajan and Lee, 2004; Dorn and Krinsley, 2011; Goldsmith et al., 2014). Varnish often consists of layers with different manganese-to-iron (Mn/Fe) ratios, which are a few tens up to a few hundreds of nanometers thick and resemble sedimentary features (e.g., Garvie et al., 2008; Krinsley et al., 1995; Macholdt et al., 2017a). It has been assumed that this layering results from variations in the mass fraction of Mn, which initially precipitates homogeneously in a single oxidation state. The oxidation state of Mn could have changed subsequent to its deposition, e.g., due to the presence of oxidizable iron species (i.e., $Fe^{2+}$) or organic matter, since Mn oxyhydroxides are known to be amongst the strongest occurring natural oxidizers and element scavengers (Tebo et al., 2005). Another process potentially reducing $Mn^{4+}$ is photoreduction, supported by available organic

matter as electron supply. Photoreduction occurs due to narrow band gaps in all Mn oxyhydroxides (Sherman, 2005). This process effectuates the dissolution of the solid Mn oxyhydroxide minerals by the reduction of immobile $Mn^{4+}$ to $Mn^{3+}$ and further to mobile $Mn^{2+}$. In most cases the released $Mn^{2+}$ is re-adsorbed to the Mn oxyhydroxide surface instead of being released into the surroundings. Since the genesis of varnish and the precipitation process of the Mn oxyhydroxides are still under debate, the aforementioned statements remain assumptions until they can be verified experimentally.

In view of the controversy regarding the varnish genesis and the scarcity of information on the varnish microchemistry, we conducted scanning transmission X-ray microscopy with near-edge X-ray absorption fine-structure spectroscopy (STXM-NEXAFS) measurements to investigate element distributions within the varnish coatings, along with spectroscopic information on the elements' binding environments and oxidation states. Experimental details on STXM-NEXAFS can be found in Kilcoyne et al. (2003) and Moffet et al. (2011). Thirteen different rock varnish samples from different environments and locations worldwide, containing diverse structures and compositions, were investigated (for details, see Macholdt et al., 2015, 2017a, b). The soft X-rays in the STXM-NEXAFS analysis generate comparatively low radiation damage, provide a high penetration depth (Guttmann and Bittencourt, 2015), and allow investigating comparatively rich spectroscopic features for a variety of elements (Hitchcock, 2015). Among the accessible elements are C, N, and O – and, thus, the composition of organic matter inside the varnish – as well as the $L$ shell [TS3] absorption edges of high-Z elements, such as the varnish-relevant elements Mn and Fe (Cosmidis and Benzerara, 2014). The Mn $L_3$ and $L_2$ absorption edges (short the Mn $L_{3,2}$ edge) are located in the energy range from $\sim 635$ to $\sim 660$ eV (i.e., electron binding energies in elemental Mn: 638.7 eV at $L_3$ and 649.9 eV at $L_2$ according to Fuggle and Mårtensson, 1980). The $L_3$ and $L_2$ edges consist of multiplets of peaks, which reflect the density of unoccupied 3d states (Gilbert et al., 2003). It is well documented in the literature that the NEXAFS spectra show different spectral patterns for the oxidation states $Mn^{2+}$, $Mn^{3+}$, and $Mn^{4+}$ (Cramer et al., 1991; Pecher et al., 2003; Gilbert et al., 2003; Nesbitt and Banerjee, 1998) and that the ratio of the $L_3$ and $L_2$ edge intensities can be taken as a measure of the 3d occupancy and thus of the valence state (Cramer et al., 1991; Kurata and Colliex, 1993). The energies of the most intense peaks within the $L_3$ multiplets for the individual oxidation states are the following: $Mn^{2+} \sim 640.2$ eV; $Mn^{3+} \sim 642.2$ eV; $Mn^{4+} \sim 643.2$ eV (Gilbert et al., 2003).

For X-ray microspectroscopic analysis, the samples must be thin enough to ensure sufficient X-ray photon transmission (Cosmidis and Benzerara, 2014). Such ultrathin samples can be prepared by focused ion beam (FIB) milling, for example by gallium ion ($Ga^+$) sputtering (Wirth, 2004; Volkert

and Minor, 2007)[1]. The FIB preparation provides several experimental advantages, such as efficient and flexible thinning of the slices as well as no risk of carbon contamination. However, several drawbacks have also been described, of which the ion-beam-related damaging of samples is the most significant.

When an accelerated ion interacts with a material, it loses its kinetic energy either via electronic (inelastic scattering of electrons) or nuclear processes (elastic collisions). Both pathways are relevant in ion milling, but nuclear processes play the predominant role (Ishitani and Kaga, 1995; Prenitzer et al., 2003). Besides the sputtering of material, it is well-known that FIB milling also implants $Ga^+$ ions in a surface-near layer (Balcells et al., 2008; Cairney et al., 2000; Prenitzer et al., 1998; Rubanov and Munroe, 2001) and thus creates Ga-rich phases with Ga fractions of up to 20 wt % in the damage layer (Susnitzky and Johnson, 1998) and up to 30 wt % in the redeposition layer (Rajsiri et al., 2002), which can melt at low temperatures (Li and Liu, 2017). The sputtered material might redeposit on the surface in some cases (Rajsiri et al., 2002). Along with the $Ga^+$ ion implantation, an amorphous damage layer is created at the surface, due to the high energy of the ion collisions (Bassim et al., 2012; Mardinly and Susnitzky, 1998; Siemons et al., 2014). In silicon, this amorphous film is typically 20–30 nm thick when using 30 keV $Ga^+$ ions (Giannuzzi et al., 2005; Rubanov and Munroe, 2004); however, even a layer as thick as 80 nm at 25 keV has been observed by Prenitzer et al. (2003). By reducing the FIB beam acceleration voltage from 30 to 6–10 kV the thickness of the amorphous layer can be reduced by a factor of 2 (Jamison et al., 2000; Rubanov and Munroe, 2004). At low FIB energies, the Ga-enriched region can have a spatial extent beyond the amorphous layer depth (Moberly-chan et al., 2007).

In addition, microstructural modifications from FIB milling of metals and ceramics have been reported, e.g., for Cu (Michael, 2006) and in manganite thin films (Balcells et al., 2008; Pallecchi et al., 2008). FIB preparations can result in reduced crystallinity of the material by generating point defects such as vacancies, interstitials, and antisite defects, due to charging effects (Li and Liu, 2017; Siemons et al., 2014), and these defects can even exceed the $Ga^+$ ion implantation depths, e.g., in $BiFeO_3$ (Siemons et al., 2014).

The amorphization coincides with the heating of the outer sample surface in the course of the collision cascade (Volkert and Minor, 2007; Fischione et al., 2017) along with the occurrence of so-called "thermal spikes", which can easily reach a few thousand kelvin (Ovchinnikov et al., 2015). Be-

---

[1]Sputtering is the physical process behind ion beam milling and means an erosion of a target material due to the transfer of kinetic energy and momentum from accelerated particles to the target material, which leads to a subsequent ejection of surface atoms in the course of a collision cascade (Wirth, 2004; Volkert and Minor, 2007).

cause of the immediate vaporization of the affected volume within $10^{-12}$ s (Ovchinnikov et al., 2015), the thermal effect on the bulk material is rather low. For samples with both good thermal conductivity and good thermal connection, ion beam heating plays a negligible role in the bulk (Volkert and Minor, 2007), but for materials with inefficient heat dissipation (e.g., due to low thermal conductivity, such as in $SiO_2$) a temperature increase up to $500\,°C$ was calculated (Ishitani and Kaga, 1995). If samples are very thin, even in metals, temperatures up to $370\,°C$ (Kim and Carpenter, 1987) or even $>400\,°C$ (Cen and Van Benthem, 2018) can be reached. This is especially destructive for organic samples, such as polymers, which tend to melt and decompose under common FIB conditions (Volkert and Minor; 2007, Schmied et al., 2014). For instance, a temperature rise of $171\,°C$ has been observed by Bassim et al. (2012) for polyacrylamide. During thinning with rather low beam currents of 0.23 nA, a temperature rise to above the melting temperature of Crystalbond™ 509 ($121\,°C$) has been observed by Li and Liu (2017). This heating can be reduced to a large extent with optimized scanning patterns (Schmied, 2014) or simply by cooling the sample (Fischione et al., 2017). The susceptibility for beam damage and the thickness of the amorphous layer strongly depends on the type of ion and the sputtered material. The damaged layer thickness typically ranges from a few to a few tens of nanometers (Mayer et al., 2007; Mikmekova et al., 2011). Sample heating also facilitates the occurrence of uncommon types of beam damage, such as preferential sputtering (Volkert and Minor, 2007), which occurs in materials with more than one atom species, especially if the compound can decompose chemically. In-depth information on sample heating (Kim and Carpenter, 1987; Cen and Van Benthem, 2018; Volkert and Minor, 2007; Ovchinnikov et al., 2015; Ishitani and Kaga, 1995), beam damage (Gutierrez-Urrutia, 2017; Mayer et al., 2007; Betz and Wehner, 1983; Prenitzer et al., 2003), and experimental reduction strategies of these effects (Bassim et al., 2012; Barber, 1993) can be found in the cited literature.

Beam damage in the samples might also result from interactions with the electron beam during scanning electron microscopy (SEM) observation, which is an integral part of the FIB preparation procedure. Inelastic scattering of electrons results in radiolytic processes, whereas elastically scattered electrons cause knock-on displacement of atoms in the substrate or from the surface, which is also known as electron beam sputtering in analogy to the above-mentioned ion beam sputtering (Saifullah, 2009; Egerton et al., 2004; Egerton, 2012; Jiang, 2016). Unsurprisingly, preferential sputtering of oxides by ions has been observed as well for electrons (Jiang, 2016). Organic molecules are more prone to radiolysis, for the main part induced by secondary electrons (Egerton, 2012). In the course of this process, molecules do not return into their original electronic states, but chemical bonds break, changing the molecule's structure, shifting their position, and causing a loss of crystallinity (Egerton

et al., 2004; Henderson and Glaeser, 1985). Mass loss or mass gain of organics due to polymerization by incoming or outgoing electrons might occur as well (Egerton et al., 2004). Inorganic samples are sensitive to both radiolysis and knock-on displacement. Radiolysis generally predominates in electrically insulating materials. In conducting specimens, "radiolysis is suppressed because of the high electron density" (Egerton, 2012) and "knock-on displacement is the sole damage process" (Egerton, 2012). For instance, alumina and transition metal oxides with Pauling electronegativity differences $>1.7$ are well known to decompose under oxygen loss through a radiolytic process named Knotek–Feibelman mechanism (Knotek and Feibelman, 1978; Egerton et al., 2004; Saifullah, 2009; Betz and Wehner, 1983; Pantano and Madey, 1981; Hoffman and Paterson, 1996). Radiolysis happens as well with high efficiency in almost all alkali and alkaline earth halides. As radiolysis shows a temperature dependence, its damaging potential can be minimized when the irradiated specimens are cooled (Pantano and Madey, 1981; Egerton et al., 2004; Egerton, 2012). This should not lead to the false conclusion that sample heating due to the electron beam is as significant as with ions. The temperature rise through electron exposure is not expected to exceed a few kelvin (Tokunaga et al., 2012; Holmes et al., 2000; Hoffman and Paterson, 1996). It should be noted that radiolytic processes come mainly into effect at the immediate surface of the sample because atom desorption is much more likely to happen here. Electron-stimulated desorption (ESD) is therefore a well-established synonym for this effect (Pantano and Madey, 1981; Egerton et al., 2004).

In this study, we summarize our observations on FIB-related damage observed during the X-ray microspectroscopic investigation of the Mn oxidation states in several rock varnish samples. The observed beam damage patterns were largely independent of the type or origin of varnish examined. We illustrate our observations by means of four different varnish samples and link them subsequently to the varnish classification scheme of Macholdt et al. (2017a).

## 2  Materials and methods

### 2.1  Focused ion beam preparation

The preparation of ultrathin slices from the rock varnish samples was performed using the lift-out FIB technique. This slicing technique was chosen since it is relatively contamination-free (except for $Ga^+$ ion implantation) and relatively fast and allows a precise selection of the preparation target area independent of the nature of the sample material or combination of materials (Mayer et al., 2007; Siemons et al., 2014). The preparation was performed at the Max Planck Institute for Polymer Research, Mainz, Germany, using an FEI Nova600Nanolab FIB dual-beam instrument (ThermoFisher Inc.). Milling was done by $Ga^+$ ion

sputtering with a resolution of $\sim 10$ nm. The FIB preparation procedure includes the following steps.

1. Before introduction of the rock varnish samples into the FIB instrument, the entire stone surface was sputter coated with 50 nm of platinum (Pt) using a Baltec MED020 sputtering equipment. The thin Pt coating makes the sample electrically conductive and thus reduces charging effects and sample shift effects.

2. In addition, the preparation site was coated within the FIB instrument with an additional, 2–3 μm thick protective Pt stripe ($50 \times 3 \, \mu m^2$) using beam-induced Pt deposition from a metallo-organic precursor gas (1 nA at 30 kV for 15 min). The Pt stripe acts as a mask to reduce damage from perpendicular ion collisions on the sample surface throughout the subsequent milling steps.

3. In a first rough cutting step (20 nA, 30 kV), two step-like trenches with a volume of about $45 \times 45 \times 30 \, \mu m^3$ on both sides of the Pt stripe were milled. This took more than 3 h on each side, depending on the individual sample characteristics.

4. The milling was followed by cleaning cross-section steps at lower beam currents (7 and 5 nA at 30 kV) to smooth the surfaces of the pre-thinned lamella ($\sim 1 \, \mu m$ thick), which were strongly affected by "curtaining". This step lasted for approximately 3 h in total on each side.

5. The samples were milled out, lifted out, transferred to an FIB lift-out grid CE1, and soldered with Pt to a copper post for the STXM-NEXAFS measurements.

6. A final stepwise thinning and polishing (1 and 0.5 nA at 30 kV) with a sample slightly tilted ($\sim 1°$) was conducted to produce wedge-shaped FIB slices with minimal thicknesses of about 100 nm at the top. This final thinning was performed from the top of the sample downwards, four to five times from each side, with decreasing currents. This step lasted for about 3.5 h for each side. To avoid breaking the FIB slices, several samples were not thinned completely but divided into two halves, one of which was thinned out more strongly than the other, as illustrated by means of one example in Fig. 1b.

SEM observation of the samples (with an acceleration voltage of 5 kV) took place during the entire FIB process and fulfilled three main purposes: (i) it allowed us to precisely define the site of milling on the varnish coated rock surfaces (step 2); (ii) it allowed monitoring the sample throughout the entire preparation procedure (steps 2–6); (iii) the electron beam neutralizes ions, which reduces charging and, therefore, minimizes drift effects.

## 2.2 STXM-NEXAFS measurements and data analysis

Subsequent to the preparation by FIB, the samples were studied using two X-ray microscopes. (i) The first STXM is located at beamline 5.3.2.2 of the synchrotron Advanced Light Source (ALS), Lawrence Berkeley National Laboratory, Berkeley, California (for details, see Kilcoyne et al., 2003). The associated bending magnet beamline allows measurements over an energy range from 250 to 800 eV. (ii) The second STXM, called MAXYMUS, is located at beamline UE46-PGM-2 of the synchrotron BESSY II, Helmholtz-Zentrum Berlin, Germany (for details, see Weigand, 2015). The associated undulator beamline allows measurements over an energy range from 250 to 1900 eV (Follath et al., 2010). Both instruments are equipped with high-energy-resolving gratings (resolving power at the carbon K edge: ALS $E / \Delta E \leq 5000$; BESSY II: $E / \Delta E \leq 8000$), a Fresnel zone plate providing a spatial resolution of about 40 nm, and phosphor-coated Lucite photomultiplier tubes for the detection of transmitted photons. At the ALS, the measurement chamber is filled with helium prior to measuring, whereas at BESSY II the measurements are conducted in a vacuum.

Measurements at both instruments are based on soft X-ray analytics, imposing less beam damage on the samples than comparable techniques, such as transmission electron microscopy with electron energy loss spectroscopy (TEM-EELS). For energy calibration, the characteristic $\pi$ resonance peak at 285.2 eV was measured on polystyrene latex (PSL) spheres prior to each measurement session. As Mn reference materials, $Mn(acac)_2$, $Mn(acac)_3$ (acac: acetylacetonate), and $MnO_2$ were used. The Mn salts were purchased from Sigma Aldrich (St Louis, USA). For the purpose of this study, STXM image "stacks" were analyzed in detail. As a routine measurement protocol, we recorded image stacks either on the entire varnish FIB slice or (with higher resolution) on specific regions of interest. The stacks typically covered the energy range from $\sim 270$ to $\sim 750$ eV (sometimes even further) and, thus, included the absorption edges of the elements carbon, potassium, calcium, nitrogen, titanium, oxygen, manganese, and iron. The Mn $L_{3,2}$ absorption edge is of primary relevance for this study.

The STXM-NEXAFS data analysis was conducted using the Interactive Data Language (IDL) widget "Analysis of X-ray microscopy Images and Spectra" (aXis2000) (Hitchcock et al., 2018), the Multivariate ANalysis Tool for Spectromicroscopy software (MANTiS-2.1.02) (Lerotic et al., 2004, 2005, 2014), as well as several custom-made software tools, programmed in Python 3.6.5. To analyze the spatial distribution of different Mn oxidation states in the FIB slices, the STXM image stacks were analyzed by a $k$-means cluster analysis with Euclidian distances. The analysis sequence included the following specific steps.

1. A careful alignment of the images in the stack was conducted with the help of a custom-made alignment tool.

2. For the subsequent analysis steps, the energy range was limited in MANTiS from 630 to 665 eV, which covers the Mn $L_{3,2}$ absorption edge. Note that in the plots of this study the energy range from 631 to 664 eV is shown.

3. According to Beer–Lambert's law,

$$OD(E) = -ln\left(\frac{I(E)}{I_0(E)}\right) = \mu(E)\rho d, \quad (1)$$

with $E$ being the X-ray photon energy, $OD(E)$ the optical density of varnish sample at given $E$, $I(E)$ the photon flux at given $E$ through the sample, $I_0(E)$ the incident photon flux at given $E$ through a sample-free region, $\mu(E)$ the energy-dependent mass absorption coefficient (see Henke et al., 1993), $\rho$ the density of absorbing atoms in the sample, and $d$ the sample thickness. The background $I_0(E)$ spectrum was determined to convert all the stack data into $OD(E)$. A modified version of the histogram-based background selection routine in MANTiS was used here.

4. An OD filter was applied to exclude pixels with $OD > 2.5$ from the analysis, which are well outside the linear regime of Beer–Lambert's law.

5. For every pixel, the Mn pre-edge value $OD_{pre}$ (averaged between 630 and 636 eV) was subtracted from the pixel-specific spectrum. Depending on the energy resolution of the stacks, data from 3 to 20 images were averaged here.

6. For every pixel, the step-function-like absorption edge was subtracted from the spectral signature. The generalized logistic function, also known as Richards' curve (Richards, 1959), was used: TS4

$$OD(E)_{\text{no-edge}} = OD(E) \quad (2)$$
$$+ \frac{OD_{post}}{\left[1 + \exp\left(-OD(E) + 0.5 \cdot OD_{post}\right)\right]^{1/(25 \cdot OD(E))}},$$

with $OD(E)_{\text{no-edge}}$ being the optical density of varnish sample at given $E$ after subtraction of the absorption edge OD from $OD(E)$, here represented by a logistic function, and $OD_{post}$ the optical density at Mn post edge, averaged between 660 and 665 eV. The pre-factor 0.5 ensures that the inflection point of the curve is located at half the edge height. The pre-factor 25 determines the steepness and symmetry of the curve. This value was found empirically and worked well for the current application. Note that without prior subtraction of $OD_{pre}$ (step 5), the following modified version of the Eq. (2) would be relevant:

$$OD(E)_{\text{new}} = OD(E) - OD_{pre} \quad (3)$$
$$+ \frac{OD_{post} - OD_{pre}}{\left[1 + \exp\left(-OD(E) + 0.5 \cdot OD_{post}\right)\right]^{1/(25 \cdot OD(E))}}.$$

7. After the preprocessing steps 1 to 4, as well as normalization steps 5 and 6, the pixels were pre-classified by means of principle component analysis (PCA) as implemented in MANTiS.

8. With the PCA results as start values, the MANTiS $k$-means cluster analysis was applied. For the current analysis $k = 4$ was chosen since it represents the smallest $k$ that still covers the observed spectral variability in the samples. Within MANTiS the "reduce thickness effect" box was checked to exclude the first PCA component in the subsequent cluster analysis (Lerotic et al., 2004), which is roughly equal to the total Mn absorption per pixel in the observed energy range. The normalization steps 5 and 6 in combination with exclusion of PCA component $s = 1$ ensure that the cluster analysis partitions the pixel spectra neither by physical thickness of the FIB slice nor by the heterogeneities in Mn distribution and $\rho_{Mn}$ but only by the spectral patterns at the Mn $L_{3,2}$ edge, which can be related to Mn oxidation states (Gilbert et al., 2003).

For the further analysis steps beyond the cluster analysis, the nonnegative matrix approximation (NNMA) routine, as implemented in MANTiS, was used to extract spectral features while constraining the weightings to be nonnegative (for details, see Mak et al., 2014). In this work, the NNMA allowed obtaining relative fractions of $Mn^{2+}$ and $Mn^{4+}$ in every pixel of the stack. Within the MANTiS NNMA routine, we used the following settings: cluster analysis output ($k = 4$) as input for NNMA; spectra similarity – 15; smoothness – 0; sparseness – 0.05; iterations – 500.

## 3 Results and discussion

The SEM overview images and STXM Mn maps in Fig. 1, which illustrate the coating thickness, morphology, and heterogeneity of the selected varnish samples, provide the context for the regions of interest that were analyzed spectroscopically (Fig. 2). The FIB slices were prepared with a wedge-like shape as illustrated in Fig. 1a2 and 1b2. Dedicated SEM measurements with a perpendicular view on the tip of the wedge showed that the thinnest part is $\sim 100$ nm thick, whereas the thickest part measures $\sim 1$ µm. For certain samples, part of the wedge was thinned out even further, allowing a comparative analysis of thicker vs. thinner wedges within the same slice (see Fig. 1b1–3).

In the course of our STXM-NEXAFS analysis of various varnish samples from different locations worldwide (see Macholdt et al., 2017a), we observed clear indications of beam-related changes in the sample composition. Specifically, differences in the spectral patterns at the Mn $L_{3,2}$ edge indicate that a beam-related reduction of the Mn oxyhydroxides has occurred. For the cluster analysis used to discriminate between these spectral patterns at the Mn $L_{3,2}$ edge – which

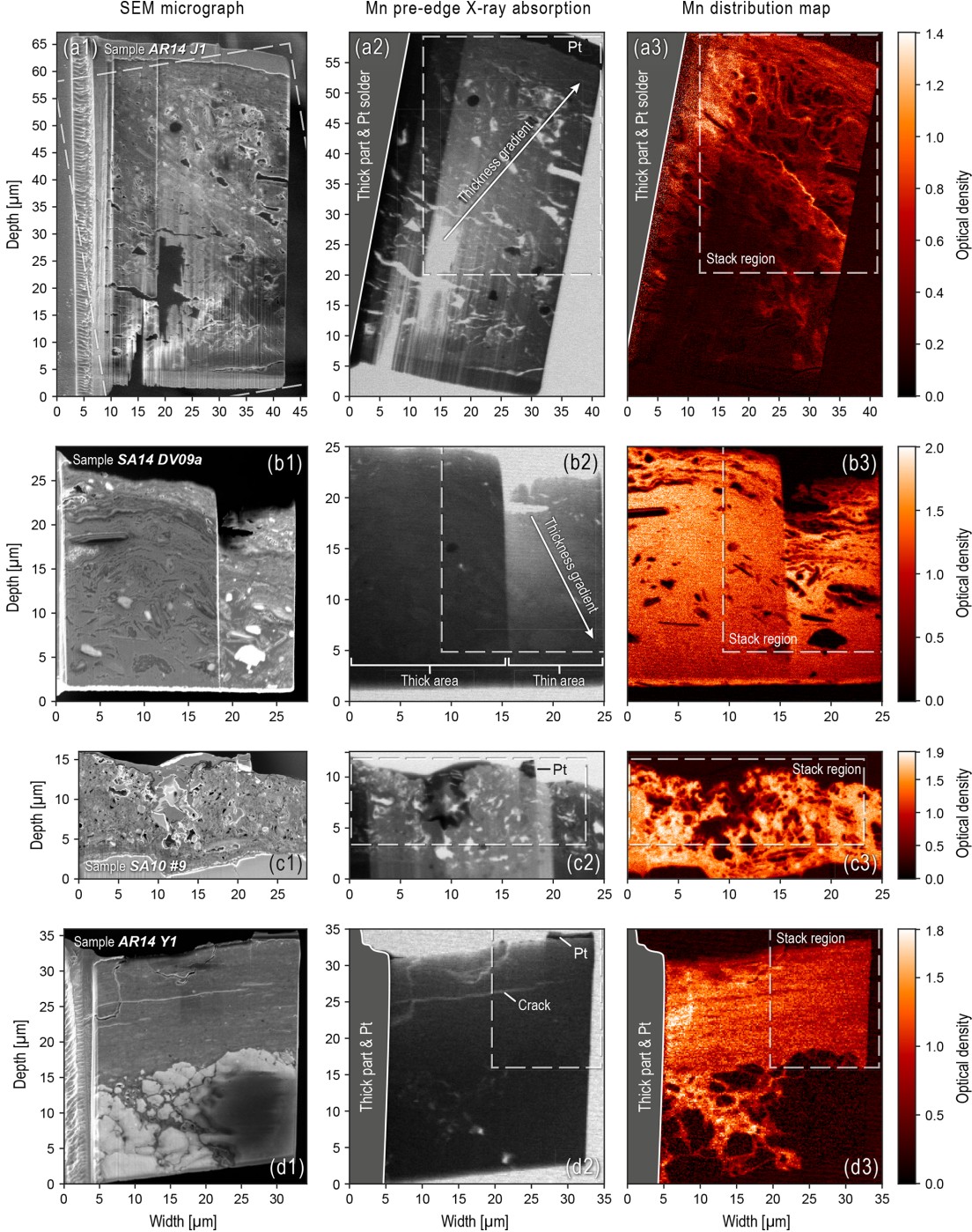

**Figure 1.** SEM images **(a1–d1)**, Mn pre-edge STXM images **(a2–d2)**, and Mn STXM maps **(a3–d3)** of FIB slices of the rock varnish samples AR14 J1 **(a1–3)**, SA14 DV09a **(b1–3)**, SA10 #9 **(c1–3)**, and AR14 Y1 **(d1–3)**. All samples are oriented such that the sample support with the Pt solder is on the left side and the rock surface at the top. The varnish layer is visible in the upper part of the images and the bedrock in the lower part in **(c1–3)** and **(d1–3)**. In panels **(c)** and **(b)**, the Mn-rich varnish layer spans across the whole FIB slice. The luminance values represent transmittance and are to some degree proportional to the sample thickness. Although this is only true of a homogeneous sample, differences in thickness due to the sample preparation and the curtaining effect are obvious and, in some cases, indicated by arrows pointing towards thicker areas in **(a2)** and **(b2)** and in addition by brackets in **(b2)**. The Mn spectra shown in Fig. 2 were collected within the dashed regions highlighted as "stack region" in the middle and right-hand columns. The Mn pre-edge images were obtained at 635 eV photon energy. The Mn maps were calculated from single images at 635 eV (pre-edge) and 643 eV (on-edge) photon energy.

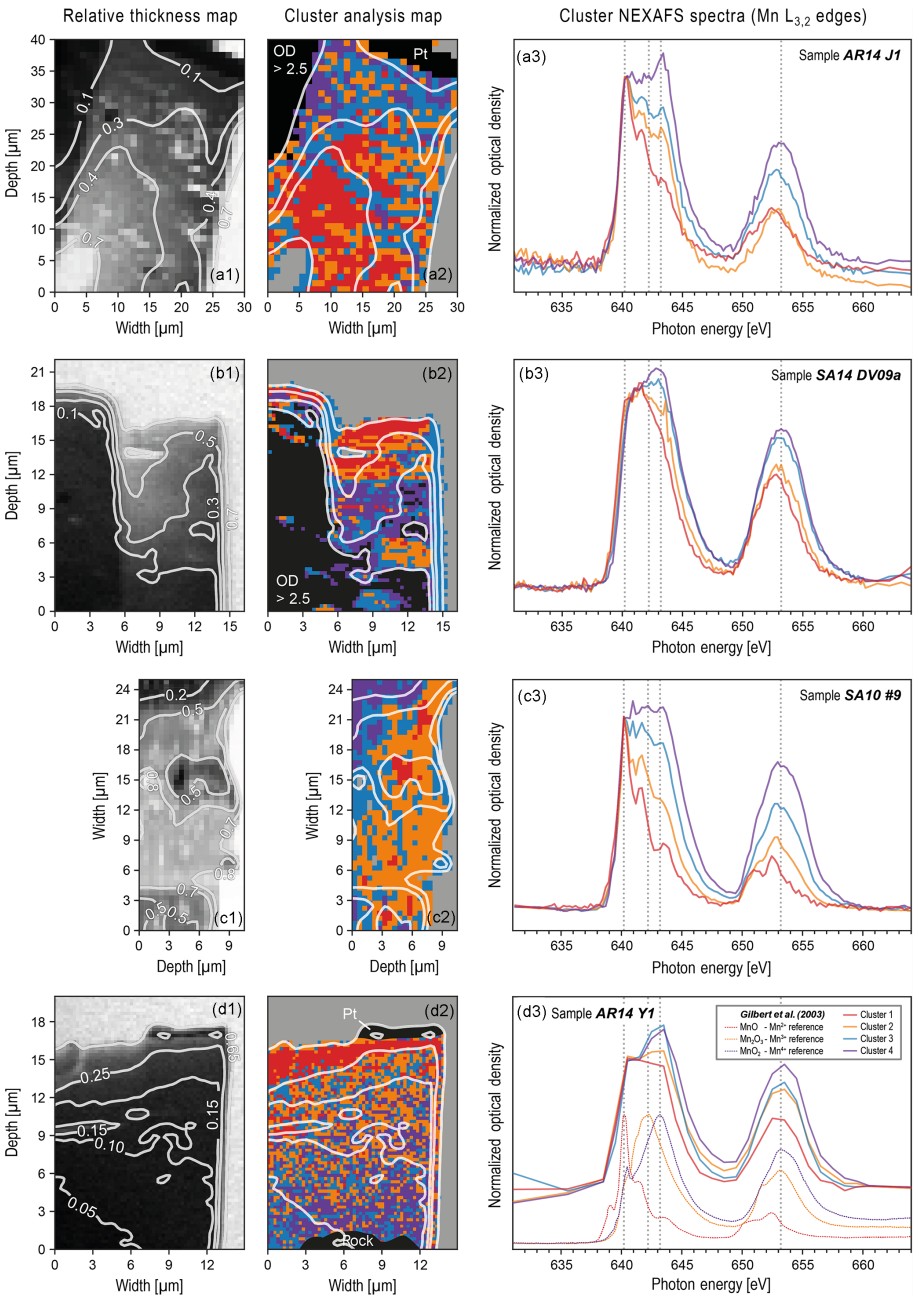

**Figure 2.** Results from *k*-means cluster analysis applied to STXM image stacks of four different varnish samples, showing the influence of beam damage (i.e., reduction of Mn oxyhydroxides) as a function of FIB slice thickness. For background information on the selected varnish samples AR14 J1, SA14 DV09a, SA10 #9, and AR14 Y1, refer to Macholdt et al. (2017a). Panels **(a1)** to **(d1)** show relative optical thickness maps obtained by averaging Mn pre-edge images between 630 and 636 eV. Relative optical thickness maps were normalized to a numeric range from 0 to 1 (0: lowest transmission in observed area; 1: full transmission). White contour lines have been calculated based on relative optical thickness maps. Panels **(a2)** to **(d2)** show spatial distribution of pixels across FIB slices partitioned into four clusters based on pixel-specific spectral patterns at the Mn $L_{3,2}$ absorption edge. White contour lines project relative optical thicknesses at the Mn pre-edge onto cluster maps. Black regions represent filtered pixels with OD > 2.5, bedrock, and Pt. Grey regions represent background pixels. Panels **(a3)** to **(d3)** show corresponding spectra from clustering (same cluster colors in **(a2)** to **(d2)** in **(a3)** to **(d3)**). Reference spectra for $Mn^{2+}$, $Mn^{3+}$, and $Mn^{4+}$ obtained from Gilbert et al. (2003) are shown in **(d3)**. The spectral pattern of cluster 1 (red) corresponds to the most reduced Mn species – similar to $Mn^{2+}$ – and is located mostly in thinnest parts of FIB slices. The spectral pattern of cluster 4 (purple) corresponds to the most oxidized Mn species – similar to $Mn^{4+}$ – and is located mostly in the thickest parts of the FIB slices. Clusters 2 (yellow) and 3 (cyan) represent intermediate states.

are a proxy for Mn oxidation states – it is important to eliminate any influence of the overall sample thickness as well as heterogeneous Mn distributions (e.g., layering) as outlined in Sect. 2.2. As a general trend, low-valence-state Mn species – similar to the $Mn^{2+}$ reference spectra – were observed in thinner regions of the FIB wedge, whereas more oxidized Mn species – similar to the $Mn^{4+}$ references spectra – dominate in the thicker regions. Figure 2 emphasizes those samples where the relationship between the optical thickness of the sample and the Mn oxidation state is resolved clearly. The gradient is most obvious in the example shown in Fig. 2d1–3. For certain samples, reduced Mn has also been observed around holes within the specimen (e.g., see cracks in Figs. 1d and 2d), which was first interpreted as a sign of the reduction of Mn by organics that had previously filled those cavities, especially since some cavities are lined with C-rich material (Macholdt et al., 2015). However, further observations suggest that the reduced Mn in the periphery of the holes can also be explained by a stronger beam exposure in the FIB preparation. For the sample AR14 Y1 in Fig. 2d1–3, we further conducted an NNMA analysis (see Sect. 2.2), which provides a proxy for the relative fractions of $Mn^{2+}$ and $Mn^{4+}$ in every pixel. This particular sample was chosen because it has the most pixels, thus providing good statistics, and the most homogeneous varnish layer (i.e., no visible clay minerals, low porosity). In Fig. 3, the scatterplots of the obtained $Mn^{2+}$ and $Mn^{4+}$ fractions against the relative (optical) thickness of the wedge further emphasize the gradients observed in Fig. 2, with the highest $Mn^{2+}$ fractions in the thinnest and highest $Mn^{4+}$ fractions in the thickest part.

The beam-damage-related gradient dominates the oxidation state distribution and, thus, is superimposed on the natural heterogeneity in Mn valence states in the varnish. Accordingly, the beam damage fundamentally hampers our original aim to use spatially resolved measurements of the Mn oxidation states for further insights into possible varnish genesis mechanisms. Some indications of layered structures, which may represent residues of the original distribution of Mn valence states, can be seen in Fig. 2a2 to b2, however, an interpretation of these structures is highly uncertain. The beam-related Mn reduction has been observed in many samples for which appropriate image stacks were recorded. Table 1 specifies whether a beam-damage effect was found in the analyzed samples and relates the samples to the varnish classification scheme, discriminating five varnish types, proposed by Macholdt et al. (2017a). According to this scheme, three of the samples in Fig. 2 (i.e., AR14 J1, AR14 Y1, and SA14 DV09a) belong to the arid desert varnish type I, whereas one sample (i.e., SA10 #9) belongs to the semiarid desert varnish type III. Multiple type I and type III samples confirm the observed trend. For statements on the varnish types II, IV, and V, however, our experimental basis is sparse: no STXM-NEXAFS data are available for type II varnish samples. For type IV and V, STXM stacks have been recorded; however,

the varnish coatings of the analyzed samples were too thin to identify clear gradients.

The following sections discuss which part of the preparation and analysis procedure has most likely caused the observed beam damage and further explore mechanistic pathways for the Mn reduction. Generally, the varnish samples experienced an intense ion and electron bombardment as well as high X-ray exposure in the course of the preparation and analysis. Accordingly, all applied techniques – FIB, SEM, and STXM – are in principle potential sources for the beam damage (Süzer, 2000; Bassim et al., 2012). The soft X-rays in STXM ($\sim 0.3$ to $0.7\,keV$), accelerated electrons in SEM ($\sim 2$ to $5\,keV$), and accelerated $Ga^+$ ions in FIB ($\sim 30\,kV$) are characterized by widely different energies. Moreover, their energies – and thus the potential damage – are deposited in the samples via different mechanistic pathways: soft X-rays mostly act via core electron excitation up to an ionization of the atom, followed by a relaxation and filling of the core hole vacancy with associated photon and Auger electron emissions. As stated in the introduction, accelerated electrons mostly interact with varnish-like specimens via inelastic scattering, possibly causing radiolytic processes in the course of electronic excitations. Accelerated ions mostly act via nuclear, i.e., elastic collisions, resulting in sputtering, but electronic excitations should not be neglected. Our experiments showed, however, that the damaging effect of STXM is negligible: in dedicated tests, sequences of successive stack scans were recorded on the same area and no difference in the spectral patterns (i.e., at the absorption edges of Mn and other elements) could be observed. Moreover, previous X-ray microspectroscopy measurements have been successfully performed on materials with different Mn oxidation states (e.g., Bargar et al., 2001; Glasauer et al., 2006; Pecher et al., 2000, 2003; Tebo et al., 2004; Toner et al., 2005).

The electrons in SEM analysis have a comparatively large penetration depth in materials due to their high velocity and small size compared with accelerated ions, such as $Ga^+$ (Mikmekova et al., 2011; Ohya and Ishitani, 2002; Prenitzer et al., 2003; Bassim et al., 2012; Pantano and Madey, 1981). Beam damage from SEM observation is particularly strong in organic matter (Bassim et al., 2012; Egerton et al., 2004). However, it seems unlikely that the $5\,keV$ electron beam from the SEM alone caused the damage visible in our STXM-NEXAFS analyses for the following reasons. (i) The critical dose for radiolytic processes in most inorganic materials (e.g., the Knotek–Feibelman mechanism in oxides), defined as the dose of $10\,\%$ compositional change of a species, is of the order of $10^{-3}$ to $10^{-2}\,C\,cm^{-2}$ (Pantano and Madey, 1981) and thus well within reach of our SEM-assisted FIB preparation procedure. However, the escape of Auger electrons, which is an essential part of the Knotek–Feibelman process along with the desorption of $O^+$ ions from the bulk, is confined within a shallow depth. For various oxides, the most probable secondary electron escape depths have been calculated by Kanaya et al. (1978), reaching a maximum at

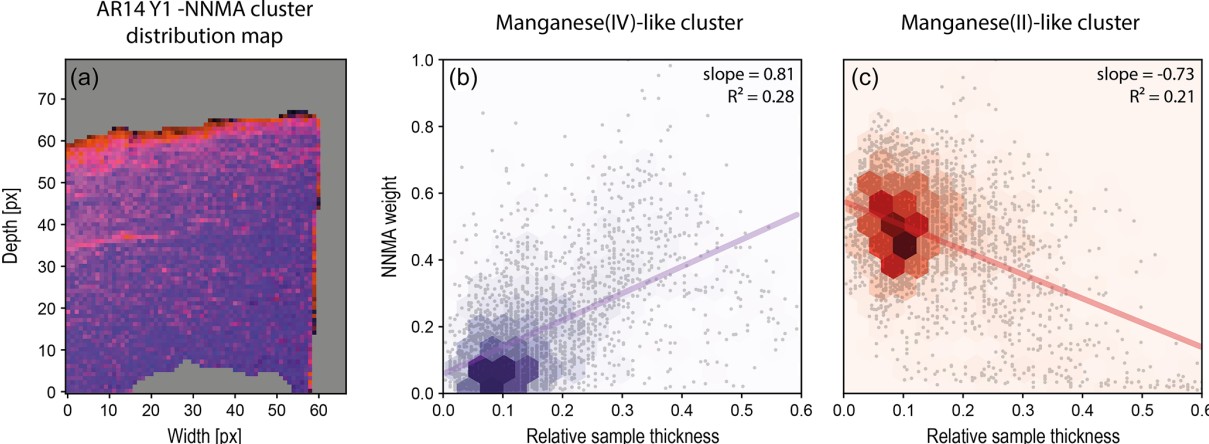

**Figure 3.** Nonnegative matrix approximation (NNMA) of sample AR14 Y1. Panel **(a)** shows the spatial distribution of the $Mn^{2+}$-associated (most reduced) cluster (red) and the $Mn^{4+}$-associated (most oxidized) cluster (violet). Panels **(b)** and **(c)** show the pixel weights from the NNMA analysis of each cluster against the "relative thickness" represented by the Mn pre-edge luminance values, scaled from 1 to 0, where 0 represents full transmission and 1 stands for the darkest pixel in the observed area. Thick areas with a relative thickness > 0.6 were masked, as were non-varnish regions (background, Pt, or rock). Compare with Figs. 1d1–3 and 2d1–3. There is a positive correlation with thickness for the more oxidized cluster and an anticorrelation for the more reduced cluster. Linear fits along with their slopes and $R^2$ values are shown to emphasize the observed trend. The increased weight of $Mn^{2+}$-like pixels in thin regions at the top, along the crack (compare with Fig. 1d2), and along the outer cutting line (right side), is very striking.

**Table 1.** Overview of all varnish samples analyzed by STXM-NEXAFS stacks, specifying whether beam-related Mn reduction was observed. Samples are grouped according to the classification scheme of Macholdt et al. (2017a): type I – arid desert varnish; type II – semiarid desert varnish; type III– semiarid desert varnish; type IV – urban area varnish; type V – river splash zone varnish.

| Sample name* | Varnish type | Beam-damage-related Mn reduction observed | Comment |
|---|---|---|---|
| AR14 J1 (AR-J) | I | Yes, clearly | Refer to Figs. 1 and 2 |
| AR14 Y1 (AR-Y) | I | Yes, clearly | Refer to Figs. 1, 2, and 3 |
| CA14 DV11 (CA-DV) | I | Unknown | Spectral gradient overlaps with porous regions, dominant varnish–rock boundary zone |
| CA14 JC8 (CA-JC) | I | Yes, clearly | – |
| IS13 V1 (IS) | I | Unknown | Low spectral quality and varnish coating too narrow |
| IS13 V3 (IS) | I | Unknown | If yes, superimposed by sample layering |
| – | II | – | No STXM-NEXAFS data for type II available |
| SA10 #9 (SA-1) | III | Yes, clearly | Refer to Figs. 1 and 2 |
| SA13 mM-f (SA-1) | III | Unknown | Sample too thick in most parts, remaining pixels indicate rather a random species distribution |
| SA14 DV09a (SA-2) | III | Yes, clearly | Refer to Figs. 1 and 2 |
| SA14 DV09b (SA-2) | III | Yes, but uncertain | Sample too thick in most parts and low statistics |
| SC | IV | Unknown | Varnish coating too narrow to resolve gradients |
| FM | IV | Unknown | Varnish coating too narrow to resolve gradients |
| E Canal | V | Unknown | Varnish coating too narrow to resolve gradients |

* Sample names kept consistent with Table 1 in Macholdt et al. (2017a).

$\sim 8$ nm for ZnO and BaO. In a worst case scenario, where 100 % $Mn^{4+}$ is converted to $Mn^{2+}$, this would mean a maximum of 16 % damaged volume at the top and 1.6 % at the bottom of our FIB slices because of the double-sided polishing. By looking at the sample NEXAFS spectra and the reference spectra in Fig. 2, it is obvious that the damaged layer we observe here is considerably thicker than that. Furthermore, part of the expected damaged layer is abraded during the final polishing phase. Moreover, Pantano and Madey (1981) showed for soda–lime–silicate glass that intermittent electron exposure, like during the SEM's scanning movement, is potentially less harmful compared to continuous exposure. In view of these aspects, the SEM's overall contribution to radiolytic damage is probably minor. (ii) The accelerating voltages necessary for knock-on displacement (electron beam sputtering) are much higher than 5 keV (Egerton et al., 2004). Therefore, a preferential sputtering of manganese oxides or in our case oxyhydroxides by the electron beam can likely be excluded. (iii) A solely SEM-induced thermal decomposition or desorption seems unlikely because electrons contribute only little to sample heating as stated before. Moreover, rock varnish is well known to have been already exposed to high temperatures in desert environments, such as up to 57 °C in Death Valley (Roof and Callagan, 2003), which is significantly beyond the temperature rise expected from electron beam exposure.

We cannot fully exclude a contribution of the electron beam to radiolytic damaging processes, however. In case of significant sample heating, which is likely to occur during the FIB treatment, as stated in the introduction, electron-induced radiolytic processes will speed up as well and might become relevant.

Considering the exceptional porosity and fragility of the analyzed samples, which can be seen from Fig. 1, thermal stress due to low thermal conductivity is in our opinion one key aspect when discussing damaging effects. The more a sample region is being thinned, the worse the heat can be dissipated, in particular at edges and around holes. We therefore assume that ion bombardment during FIB preparation is at least an essential ingredient for, if not the main contributor to, the observed changes in Mn valence states for two reasons: (i) the high kinetic energy (Sezen et al., 2011) of the $Ga^+$ ions ($\sim 5700$ times higher than 5 keV electrons), which is almost fully converted to heat within the sample, and (ii) the rather long ion beam exposure during milling. Very little information on ion-induced alterations of the composition of complex (e.g., geological) samples is available in the literature. However, analogies and extended literature on related effects can be found in the research fields of planetary science and materials science, which are helpful for the interpretation of our observations. Accordingly, the following sections provide a literature synthesis as a basis for further discussion of our results.

Chemical reduction by means of an electronic process should in principle play a minor role in ion beam milling,

as stated in the introduction. But evidence for effective "bombardment-induced decomposition" (Kelly, 1989) and reduction can even be found in outer space. The low albedo of silicate rocks from the moon's surface was attributed to solar wind bombardment (Hapke, 1973), which causes an enrichment of nanoscale metallic iron particles in the near-surface layer of lunar regolith via a preferential sputtering mechanism, where oxygen is preferentially sputtered off, leaving the reduced bare metal behind (Betz and Wehner, 1983). Whether micrometeoroid impacts have a major contribution to this so-called lunar "space weathering" by (re)melting the surface layer of lunar soil grains or whether solar wind contributes more due to ion (H and He) implantation is still under discussion, however (Kuhlman et al., 2015; Pieters and Noble, 2016). Note that Christoffersen et al. (2012) tried to simulate space weathering by $Ga^+$ ion FIB milling but without success. In their study, the irradiation only lasted 25 min, however, compared with many hours of FIB milling from both sides in our sample preparation procedure. Besides that, we do not want to make oversimplifications by directly comparing lunar pyroxene reduction to elemental iron with desert varnish Mn species reduction to $Mn^{2+}$.

In materials science, the sputter reduction by ions during milling is well known, especially in metal oxides (Hofmann and Sanz, 1983; Betz and Wehner, 1983; Mitchell et al., 1990; Parker and Kelly, 1973; McIntyre and Zetaruk, 1977; Fondell et al., 2018). Hydroxides are reduced upon ion bombardment as well, as was shown for $Ni(OH)_2$, $Co(OH)_2$, and FeOOH by Chuang et al. (1978). The associated oxygen loss was often attributed to preferential sputtering (Malherbe et al., 1986; Mitchell et al., 1990; Parker and Kelly, 1973; Betz and Wehner, 1983) "triggered by differences of mass, chemical binding or volatility" (Kelly, 1989). The decomposition caused by volatility differences, known as thermal sputtering was "found to explain many examples of oxygen loss from oxides" (Kelly, 1989). However, in many cases there is no single mechanism responsible for the observed decomposition, but rather there is interplay between several chemical and physical effects (Kelly, 1989; Mitchell et al., 1990; Barber, 1993). In the aforementioned studies, the oxides or hydroxides were either grown as films on a substrate or on macroscopically thick bulk materials and thus thermally rather well connected. The observed thicknesses of the damage layer typically range from a few nanometers (Mitchell et al., 1990; Hofmann and Sanz, 1983) to a few tens of nanometers (Betz and Wehner, 1983), although for $MoO_3$ an altered layer of approximately 115 nm thickness has been observed (Naguib and Kelly, 1972). The oxygen-depleted layer thicknesses are therefore in the same size range as the surface amorphization. Our samples are only 100 nm thick at the top and were ion-milled from both sides, although final thinning and polishing only happened under grazing irradiation conditions. The implication of different impact angles is mentioned by Prenitzer et al. (2003) and Mikmekova et

al. (2011). While it is true that the penetration depths and therefore the damaged volumes become generally smaller under grazing incidence, the complex sample morphology and composition in our case adds a large uncertainty to the real ion impact angles and the depth of penetration.

During our in-depth literature search for samples comparable to rock varnish, we came across the recent study on ion bombardment of iron oxide thin films by Fondell et al. (2018). They report sputter reduction to FeO under remarkably soft conditions (200 V, $Ar^+$ ions) after just 8 min of sputtering. Only very few data on Mn oxides were published by Mitchell et al. (1990) along with various other oxides. Kelly (1989) states in his comprehensive overview that Mn oxides are not reduced further than to MnO, which is thermodynamically stable at least to temperatures up to 900 K. This agrees well with our observations. The spectral signatures in Fig. 2a3–d3 approach the $Mn^{2+}$ spectral pattern in the highly thinned regions. In consideration of the findings by Fondell et al. (2018), a similar behavior of oxidic Mn compounds can be reasonably assumed and due to the comparably harsh milling parameters we used, a more extensive reduction at greater depths seems plausible.

## 4 Conclusions

In this study, we found that the Mn oxidation states investigated using STXM-NEXAFS were modified during the sample preparation procedure by reducing $Mn^{4+}$ to $Mn^{2+}$. Overall, we found that artifacts are produced during the preparation of the samples by FIB and monitoring by SEM, which creates a high degree of uncertainty for oxidation state analyses. This study supplies a clearer picture on the type of artifacts created, providing the possibility to address these issues carefully in follow-up studies. Although we were not able to confirm oxidation state alteration in all the samples given in Table 1, we see no reasons why varnish types II, IV, and V, from which no suitable samples were available, should behave differently.

From an overview of the existing literature we further conclude the following points. (i) Ion-bombardment-induced reduction in multicomponent specimens is a common phenomenon. The effect has been extensively discussed by Kelly (1989). (ii) Ion beam heating of thermally low-conducting specimens plays a significant role therein because it facilitates preferential sputtering (i.e., thermal sputtering) as well as chemical decomposition reactions (i.e., radiolysis). For instance, Fondell et al. (2018) report that sputter reduction of maghemite is similar to heat treatment of the sample in vacuum. The dissociation of carbonates (Christie et al., 1981) and sulfates (Contarini and Rabalais, 1985) has been observed and attributed to a combination of thermal sputtering connected to thermal spikes and electron sputtering. Momentum transfer alone (preferential sputtering due to mass differences) could not explain the observed reactions. A follow-up study on previously heated samples could therefore help to better understand the potential thermal decomposition of rock varnish. (iii) The expected damaged layer depths in FIB milling might have been underestimated in multicomponent samples, especially when they consist of oxidic compounds. (iv) The possible contribution of the electron beam from SEM is yet unknown. In combination with ion beam heating, electron-induced radiolytic processes might come into effect. Further damaging contribution from electrons might arise from interaction with the ions' shell electrons and from secondary electrons liberated in the course of the collision cascade.

As FIB is a widely used technique to produce ultrathin slices of rock samples, one needs to be aware of these problems and choose preparation parameters that help to keep damage to a minimum. To reduce or minimize the damaged volume, the preparation procedure could be conducted using not only low currents but lower voltages during FIB preparation. In contrast, lowering the accelerating voltage in SEM might have an opposing, more damaging effect (Joy and Joy, 1996). If available, a cryo-FIB approach (Bassim et al., 2012) could be applied. Sezen et al. (2011) showed, however, that cryogenic conditions could not prevent or even slow down the degradation of conjugated polymers during FIB milling. Alternatively, argon ion slicing (Stojic and Brenker, 2010) may be a more gentle and, therefore, suitable approach to reduce beam damage (e.g., Mn reduction) in the preparation of ultrathin varnish slices. Iodine ion milling as mentioned in Barber (1993) might be even less damaging. Fischione et al. (2017) established a method in which the damaged surface layers can be removed after FIB milling by a small-spot argon ion milling process. However, it is left to further studies to investigate whether oxidation states can indeed be kept unchanged using such more gentle preparation approaches.

*Data availability.* The STXM-NEXAFS stacks used for Figs. 2 and 3, the cluster analysis results, and the corresponding spectra have been deposited in Edmond, the Max Planck society's open-access data repository under https://doi.org/XX.XXXXX/X.XX TS5. For specific data requests, please contact the corresponding authors. CE2

*Author contributions.* JDF, CP, and DSM are responsible for the data analysis, the conceptual design of the paper, the literature search, and the paper writing. CP and MOA supervised this study and the paper writing. DSM, CP, JDF, MOA, and BW performed the STXM measurements, which were conducted with the technical assistance of ALDK and MW. The FIB preparation was done by MM and MK. All authors contributed to data discussion and paper finalization. CE3

*Competing interests.* The authors declare that they have no conflict of interest.

*Acknowledgements.* This work was supported by the Max Planck Graduate Center with the Johannes Gutenberg University Mainz (MPGC), the Max Planck Society, and King Saud University. The ALS is supported by the Director, Office of Science, Office of Basic Energy Sciences, of the US Department of Energy under Contract DE-AC02-05CH11231. We thank the Helmholtz-Zentrum Berlin for the allocation of the synchrotron radiation beam time at BESSY II. We thank Ulrich Pöschl and Gerald Haug for support and stimulating discussions. We also thank Adam Hitchcock for his support providing information, literature, and constructive criticism during the review process. We further thank referee 2 for helpful comments.

The article processing charges for this open-access publication were covered by the Max Planck Society.

Edited by: Maria Genzer
Reviewed by: Adam Hitchcock and one anonymous referee

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

**Remarks from the language copy-editor**

CE1     Inserted with slight grammatical adjustment ("an FIB").
CE2     Text inserted with minor adjustments to punctuation.
CE3     Text inserted with minor adjustments to punctuation.

**Remarks from the typesetter**

TS1     Please note that it corresponds to our guidelines to separate the affiliation in this case. Otherwise we cannot see which author belongs to which department. So please make a difference in the author list. For now I added all authors to both departments. Thanks.
TS2     Thanks for your comment but it is indeed a standard to have a skinny space between number and tilde.
TS3     As L stands for a value/parameter, it is correct to show it in italic font while L-edge is a spectroscopic technique which seems to be written in roman font.
TS4     To insert your requested equation changes, we have to ask the handling editor for an approval. Please explain what you corrected in detail. Thank you.
TS5     Needs to be updated.
TS6     Please note that we do not add URLs to article citations but volume and page range are fine.