# Peer review of "Artifacts from manganese reduction in rock samples prepared by focused ion beam (FIB) slicing for X-ray microspectroscopic analysis"

_Geoscientific Instrumentation, Methods and Data Systems, 2018_

## Referee Comment (RC1) · AP Hitchcock (Referee) · 2 Jul 2018

This manuscript presents an analysis of the Mn oxidation states in a rock vanish, for one sample prepared by room temperature focused ion beam (FIB) milling. This is said to be representative of a larger set of samples which exhibit similar properties.

The data quality is good, and the analytical methods are appropriate. The authors interpret the results as evidence that the FIB is drastically alters the Mn oxidation state, which is quite plausible since metal ion reduction by FIB is known from other work.

However the conclusion is not really supported by the data. The evidence, as presented, suggests the thicker regions in the sample have lower amounts of Mn(2+), but there really is not very strong evidence for the presence of Mn(4+) (green in Fig 3), except in some very localized spots. Similarly, the areas identified as Mn(2+) (blue in Fig 3) are localized in a line across the sample, whereas, if the reduction was due to FIB beam damage, then I would expect a more broad distribution of Mn(2+) signal, more like the Mn(3+) distribution (red in fig. 3) reported by the authors.

One way to make a more convincing argument would be to use the authors' favourite method to estimate the fraction of Mn(2+), Mn(3+) and Mn(4+) at each pixel in the thinner area they say is not affected by absorption saturation, and also to estimate the thickness of each pixel (for example, from the average STXM image below the onset of the Mn2p edge). A plot of the fraction of each Mn oxidation state as a function of thickness (with suitable binning to improve statistics) should then directly reveal any (anti)-correlation of Mn(2+) amount with thickness.

A second way would be to do that type of thickness-oxidation state correlation on several other samples the authors say they have made by FIB and analysed by STXM.

Perhaps the best way might be to compare results from this sample, with the results from one where the final FIB polish was deliberately done at high keV and high current – i.e. deliberately emphasize the damage.

As is, I think the manuscript is a good contribution to the discussion on optimizing sample preparation for these types of samples. It should stimulate discussion.

In the future (NOT SUGGESTED FOR THIS PAPER), the authors might want to explore using STXM with total electron yield (TEY) detection on a cleaved or other, non-FIB prepared sample.

Detailed comments

* (p2 27) 'phosphor screen;' – change to 'phosphor coating – screen implies imaging

but only a single number is read from the detector at each pixel in a STXM image.

* (p2, 27) 'generated visible' → 'generated burst of visible' – it is not individual visible photons, but bursts since there is ∼1 visible photon per ∼3 eV of photon energy

* (p3, 16) to a few hundred degrees - is this proven or speculation ? reference ? would it be worth to compare FIB of RT and heated varnish samples ? – if the heat during FIB is important, a change in degree of radiation damage might occur. (cryo-FIB is known to reduce damage).

* (p 4 ,14) the metal coating is done, in part to reduce damage from heating or charging. Was there any study of the dependence on the amount of damage on the thickness of the Pt coating ?

* (p 6, 23) 'main absorption edges at different energies' → it would be useful to define what you mean by 'edges'. I suspect the XPS peak energies for Mn(2+/3+/4+). As you note, the spectra of each ox state (in fact each 'compound' or local environment) are characterized by multiple peaks, and a single 'energy', 'edge (in XAS sense)' or 'peak' is not enough to uniquely identify oxidation state. It is the overall pattern that is needed. This should be brought out.

* (p 6, 25) connecting multiplets to oxidation state is actually a gross simplification. Multiplet refers specifically to the [core electron – valence electron] exchange inter- action. Oxidation state (interpreted as a net valence electron count) is only indirectly connected.

* (p 6, 27) " for each oxidation state the absorption at a certain energy (Mn 2+ ∼ 639.7 eV, Mn 3+ ∼ 641.35 eV, Mn 4+ ∼ 643.05 eV) is predominant, so that the oxidation states can be distinguished from each other " again, I would stress that it is the PAT- TERN of peaks that is connected to oxidation, not a single peak.

* (p 7, caption to Fig 2) The caption calls (c) and (e) "images", but they are not – they are color coded cluster signal distributions.
* (p 8, 37) ' While STXM-NEXAFS measurements are conducted with energies in the eV range, FIB preparation and SEM imaging utilize energies in the keV range.' The correlation of damage-potential and particle energy is an oversimplification. The X-rays transfer ALL their energy to the sample on absorption, whereas the ion and electron beams transfer only a portion. For ions it is momentum rather than energy transfer that is important I suspect. The reduction is probably done by liberated electrons.

---

## Referee Comment (RC2) · Anonymous Referee #2 · 15 Oct 2018

General comment:

The paper reports on a study investigating Mn rich rock coatings to find means to identify the different mechanisms in the genesis of the crust formation. The authors have examined the oxidation states of Mn in the crusts using FIB (focused ion beam) slicing and scanning transmission X-ray microscopy. The finding of the manuscript is that preparation of specimens by FIB and monitoring of samples by SEM cause artifacts that complicate the oxidation state analysis.

[Figure]

The reader cannot avoid an impression that the manuscript is a spin-off produced by an ambitious study originally intended to resolve the profile of Mn oxidation states all the way through the varnish layer and to track the different mechanisms behind the development of the coating. However, as sample preparation techniques remains an issue within geosciences, the manuscript makes a fair contribution to this field of science.

Specific comments:

Page 8 line 32: "To verify whether a layer of modified material is actually distributed homogeneously on the surface of the sample"

Why would you assume an even distribution? Please justify.

Page 11 line 10: "As there is at this time no alternative to FIB as sample preparation technique to produce intact ultra-thin slices of rock samples,"

The reader might wonder which are the benefits provided by FIB compared to, e.g., Argon ion slicing that has been also used in production of thin foils especially for TEM. According to this statement, you don't consider Argon ion slicing as an alternative to FIB. However, if this is the case especially for the samples used in this study, the reader would appreciate some reasoning.

Page 3 line 33: "Here we report about our findings observed during the investigation of the Mn oxidation states in 14 rock varnish samples, collected in different environments and countries." and Page 4 line 2: "For the sake of brevity, and since all samples showed the same phenomena, these findings will be exemplified using measurements on one of the samples."

It is interesting that no differences between the varnishes were found especially as you have previously reported (Macholdt et al. 2017) that layers of Mn-rich material and structures like cavities vary significantly between coatings of rock samples collected from different environments and regions. Perhaps you could refer to your earlier study to emphasize the importance of the finding of this manuscript - that the sample preparation of this sort produces similar kind of artifacts no matter what the structure of the varnish is.

Page 10 line 8: "we found that artifacts are produced during the preparation of the samples by FIB and monitoring by SEM, which create a high degree of uncertainty for oxidation state analyses."

The reader would appreciate a quantitative estimate. Would it be possible to give a rough estimate on how much sample preparation of this kind adds to the total uncertainty – on the basis of the case presented in the manuscript?

---

## Author Comment (AC1) · 21 Jan 2019

We appreciate the very thorough and helpful comments by Adam Hitchcock, which have been considered carefully and helped to improve the quality of our manuscript. The referees' comments and our responses are outlined in detail below:

[1.1]    Referee comment: The authors interpret the results as evidence that the FIB is drastically alters the Mn oxidation state, which is quite plausible since metal ion reduction by FIB is known from other work. However, the conclusion is not really supported by the data. The evidence, as presented, suggests the thicker regions in the sample have lower amounts of Mn(2+), but there really is not very strong evidence for the presence of Mn(4+) (green in Fig 3), except in some very localized spots. Similarly, the areas identified as Mn(2+) (blue in Fig 3) are localized in a line across the sample, whereas, if the reduction was due to FIB beam damage, then I would expect a more broad distribution of Mn(2+) signal, more like the Mn(3+) distribution (red in fig. 3) reported by the authors. One way to make a more convincing argument would be to use the authors' favourite method to estimate the fraction of Mn(2+), Mn(3+) and Mn(4+) at each pixel in the thinner area they say is not affected by absorption saturation, and also to estimate the thickness of each pixel (for example, from the average STXM image below the onset of the Mn2p edge). A plot of the fraction of each Mn oxidation state as a function of thickness (with suitable binning to improve statistics) should then directly reveal any (anti)-correlation of Mn(2+) amount with thickness. A second way would be to do that type of thickness-oxidation state correlation on several other samples the authors say they have made by FIB and analysed by STXM.

Author Response:

The worthwhile ideas towards a clearer representation of our data gave us the occasion to extensively rework our results and apply a sample thickness normalization routine prior to the spectral clustering. Furthermore, we increased the number of discussed samples to four to provide more (statistical) weight to our argumentation. In addition, results from a total of thirteen samples have been summarized in Table 1. We explain our newly developed sample analysis procedure in the reworked sections below:

"2.2 STXM-NEXAFS measurements and data analysis

[revised manuscript text omitted]

SEM micrograph | Mn pre-edge x-ray absorption | Mn distribution map

**Fig. 1:** SEM images (**A₁-D₁**), Mn pre-edge STXM images (**A₂-D₂**), and Mn STXM maps (**A₃-D₃**) of FIslices of the rock varnish samples AR14 J1 (**A₁₋₃**), SA14 DV09a (**B₁₋₃**), SA10 #9 (**C₁₋₃**), and AR14 Y1 (**D₁₋₃**). All samples are oriented such that the sample support with the Pt solder is on the left side and the rock surface at the top. The varnish layer is visible in the upper part of the images and the bedrock in the lower part in panels C and D. In panels A and B, the Mn-rich varnish layer spans across the whole FIB slice. The luminance values represent transmittance and are to some degree proportional to the sample thickness. Although this is only true for a homogeneous sample, differences in thickness due to the sample preparation and the curtaining effect are obvious and, in some cases, indicated by arrows pointing towards thicker areas, or via brackets in column two. The Mn spectra shown in Fig. 2 were collected within the dashed regions highlighted as 'stack region' in column two and three. The Mn pre-edge images were obtained at 635 eV photon energy. The Mn maps were calculated from single images at 635 eV (pre-edge) and 643 eV (on-edge) photon energy.

Relative thickness map     Cluster analysis map     Cluster NEXAFS spectra (Mn L$_{3,2}$ edges)

[Figure]

**Fig. 2:** Results from *k*-means cluster analysis applied to STXM image stacks of four different varnish samples, showing influence of beam damage (i.e., reduction of Mn oxyhydroxides) as a function of FIB slice thickness. For background information on the selected varnish samples AR14 J1, SA14 DV09a, SA10 #9, and AR14 Y1 refer to Macholdt et al. (2017a). Panels **A₁** to **D₁** show relative optical thickness maps obtained by averaging Mn pre-edge images between 630 and 636 eV. Relative optical thickness maps were normalized to a numeric range from 0 to 1 (0 = lowest transmission in observed area; 1 = full transmission). White contour lines have been calculated based on relative optical thickness maps. Panels **A₂** to **D₂** show spatial distribution of pixels across FIB slices partitioned into four clusters based on pixel-specific spectral patterns at the Mn $L_{3,2}$ absorption edge. White contour lines project relative optical thicknesses at the Mn pre-edge onto cluster maps. Black regions represent filtered pixels with OD > 2.5, bedrock and Pt. Grey regions represent background pixels. Panels **A₃** to **D₃** show corresponding spectra from clustering (same cluster colors in **A₂** to **D₂** in **A₃** to **D₃**). Reference spectra for $Mn^{2+}$, $Mn^{3+}$ and $Mn^{4+}$ obtained from Gilbert et al. (2003) are shown in **D₃**. The spectral pattern of cluster 1 (red) corresponds to the most reduced Mn species – similar to $Mn^{2+}$ – and is located mostly in thinnest parts of FIB slices. The spectral pattern of cluster 4 (purple) corresponds to the most oxidized Mn species – similar to $Mn^{4+}$ – and is located mostly in the thickest parts of the FIB slices. Clusters 2 (yellow) and 3 (cyan) represent intermediate states.

**Table 1.** Overview of all varnish samples analyzed with appropriate STXM-NEXAFS stacks as well as with specification if beam-related Mn reduction has been observed. Samples are grouped according to classification scheme by Macholdt et al. (2017a): type I = arid desert varnish, type II = semi-arid desert varnish, type III = semi-arid desert varnish, type IV = urban area varnish, type V = river splash zone varnish.

| Sample name[1] | Varnish type | Beam damage-related Mn reduction observed: | Comment |
|---|---|---|---|
| AR14 J1 (AR-J) | I | Yes, clearly | Refer to Fig. 1 & 2 |
| AR14 Y1 (AR-Y) | I | Yes, clearly | Refer to Fig. 1, 2 & 3 |
| CA14 DV11 (CA-DV) | I | Unknown | Spectral gradient overlaps with porous regions, dominant varnish/rock boundary zone |
| CA14 JC8 (CA-JC) | I | Yes, clearly | -- |
| IS13 V1 (IS) | I | Unknown | Low spectral quality & varnish coating too narrow |
| IS13 V3 (IS) | I | Unknown | If yes, superimposed by sample layering |
| -- | II | -- | No STXM-NEXAFS data for type II available |
| SA10 #9 (SA-1) | III | Yes, clearly | Refer to Fig. 1 & 2 |
| SA13 mM-f (SA-1) | III | Unknown | Sample too thick in most parts, remaining pixels indicate rather a random species distribution |
| SA14 DV09a (SA-2) | III | Yes, clearly | Refer to Fig. 1 & 2 |
| SA14 DV09b (SA-2) | III | Yes, but uncertain | Sample too thick in most parts & low statistics |
| SC | IV | Unknown | Varnish coating too narrow to resolve gradients |
| FM | IV | Unknown | Varnish coating too narrow to resolve gradients |
| E Canal | V | Unknown | Varnish coating too narrow to resolve gradients |

[1] *Sample names kept consistent with Table 1 in Macholdt et al. (2017a)*

As suggested, the (anti-)correlation of Manganese species with thickness was exemplarily done for one sample:

> "[…] For the sample AR14 Y1 in Fig. 2D$_{1-3}$, we further conducted an NNMA analysis (see Sect. 2.2), which provides a proxy for the relative fractions of $Mn^{2+}$ and $Mn^{4+}$ in every pixel. This particular sample was chosen because it has the most pixels, thus providing good statistics, and the most homogeneous varnish layer (i.e., no visible clay minerals, low porosity). In Fig. 3, the scatter plots of the obtained $Mn^{2+}$ and $Mn^{4+}$ fractions against the relative (optical) thickness of the wedge further emphasize the gradients observed in Fig. 2, with highest $Mn^{2+}$ fractions in the thinnest and highest $Mn^{4+}$ fractions in the thickest part. […]"

[Figure]

**Fig. 3:** Non-negative matrix approximation (NNMA) of sample AR14 Y1. Panel A shows the spatial distribution of the $Mn^{2+}$-associated (most reduced) cluster (red) and the $Mn^{4+}$-associated (most oxidized) cluster (violet). Panels B and C show the pixel weights from the NNMA analysis of each cluster against the "relative thickness" represented by the Mn pre-edge luminance values, scaled from 1 to 0, where 0 represents full transmission and 1 stands for the darkest pixel in the observed area. Thick areas with relative thickness > 0.6 were masked, as well as non-varnish regions (background, Pt or rock). Compare with Fig. 1D and Fig. 2D. There is a positive correlation with thickness for the more oxidized cluster and an anti-correlation for the more reduced cluster. Linear fits along with their slopes and R²-values are shown to emphasize the observed trend. Very striking is the increased weight of $Mn^{2+}$-like pixels in thin regions at the top, along the crack (compare with Fig. 1 D$_2$) and along the outer cutting line (right side)."

[1.2]  Referee comment: (p2 27) 'phosphor screen;' – change to 'phosphor coating – screen implies imaging but only a single number is read from the detector at each pixel in a STXM image.

[1.3]  Referee comment: (p2, 27) 'generated visible'→'generated burst of visible' – it is not individual visible photons, but bursts since there is ~1 visible photon per ~3 eV of photon energy

Author Response: To make the introduction more concise, we deleted the general information on STXM, since the details are well documented in the references we are listing in the newly rephrased paragraph:

"In view of the controversy regarding the varnish genesis and the scarcity of information on the varnish microchemistry, we conducted STXM-NEXAFS measurements to investigate element distributions within the varnish coatings, along with spectroscopic information on the elements' binding environments and oxidation states. Experimental details on STXM-NEXAFS can be found in Kilcoyne et al. (2003) and Moffet et al. (2011)."

[1.4]  Referee comment: (p3, 16) to a few hundred degrees - is this proven or speculation ? reference ? would it be worth to compare FIB of RT and heated varnish samples ? – if the heat during FIB is important, a change in degree of radiation damage might occur. (cryo-FIB is known to reduce damage).

Author Response: Regarding this valuable comment, we re-evaluated the damaging potential of sample heating from SEM and FIB, added some quantitative information, and rephrased essential parts of the introduction:

[revised manuscript text omitted]

[1.5]  Referee comment: (p 4 ,14) the metal coating is done, in part to reduce damage from heating or charging. Was there any study of the dependence on the amount of damage on the thickness of the Pt coating ?

Author Response: The application of the Pt stripe is an integral part of the FIB preparation procedure and a consistently thick coating was applied to all samples to keep them comparable. The ablation of the Pt strip was also used as an indicator for a sufficiently thinned FIB slice.

In our data, compare Fig. 2 $A_2$ and $D_2$, we see no damage reducing effect due to the Pt stripe's improved heat dissipation. However, it would be interesting to test this assumption on a FIB sample without a Pt stripe. We also rephrased the sentence to which the question refers:

"The Pt stripe acts as a mask to reduce damage from perpendicular ion collisions on the sample surface throughout the subsequent milling steps."

[1.6]  Referee comment: (p 6, 23) 'main absorption edges at different energies'→it would be useful to define what you mean by 'edges'. I suspect the XPS peak energies for Mn(2+/3+/4+). As you note, the spectra of each ox state (in fact each 'compound' or local environment) are characterized by multiple peaks, and a single 'energy', 'edge (in XAS sense)' or 'peak' is not enough to uniquely identify oxidation state. It is the overall pattern that is needed. This should be brought out.

Author Response: True. See response to [1.8].

[1.7]    Referee comment: (p 6,  25) connecting multiplets to oxidation state is actually a gross simplification. Multiplet refers specifically to the [core electron – valence electron] exchange interaction.  Oxidation state (interpreted as a net valence electron count) is only indirectly connected.

Author Response: Agree. See response to [1.8].

[1.8]    Referee comment: (p 6, 27) "for each oxidation state the absorption at a certain energy (Mn 2+ ~ 639.7 eV, Mn 3+ ~641.35 eV, Mn 4+ ~ 643.05 eV) is predominant, so that the oxidation states can be distinguished from each other" again, I would stress that it is the PATTERN of peaks that is connected to oxidation, not a single peak.

Author Response to [1.6], [1.7], and [1.8]: We agree with the referee that the wording in the corresponding paragraph is confusing. Accordingly, we reworked the original text section

> "[…] The Mn $L_3$ and $L_2$ absorption edges (short the Mn $L_{3,2}$ edge) are located in the energy range from ~635 to ~660 eV (i.e., electron binding energies in elemental Mn: 638.7 eV at $L_3$ and 649.9 eV at $L_2$ according to Fuggle and Mårtensson, 1980). The $L_3$ and $L_2$ edges consist of multiplets of peaks, which reflect the density of unoccupied 3d states (Gilbert et al., 2003). It is well documented in the literature that the NEXAFS spectra show different spectral patterns for the oxidation states $Mn^{2+}$, $Mn^{3+}$, and $Mn^{4+}$ (Cramer et al., 1991, Pecher et al., 2003, Gilbert et al., 2003, Nesbitt and Banerjee, 1998) and that the ratio of the $L_3$ and $L_2$ edge intensities can be taken as a measure for the 3d occupancy and thus for the valence state (Cramer et al., 1991, Kurata and Colliex, 1993). The energies of the most intense peaks within the $L_3$ multiplets for the individual oxidation states are the following: $Mn^{2+}$ ~ 640.2 eV, $Mn^{3+}$ ~ 642.2 eV, $Mn^{4+}$ ~ 643.2 eV (Gilbert et al., 2003)."

and added a few words of explanation to the 'results and discussion' section:

> "Specifically, differences in the spectral patterns at the Mn $L_{3,2}$ edge indicate that a beam-related reduction of the Mn oxyhydroxides has occurred. For the cluster analysis used to discriminate these spectral patterns at the Mn $L_{3,2}$ edge – which are a proxy for Mn oxidation states –, it is important to eliminate any influence of the overall sample thickness as well as heterogeneous Mn distributions (e.g., layering) as outlined in Sect. 2.2."

[1.9]    Referee comment: (p 7, caption to Fig 2) The caption calls (c) and (e) "images", but they are not – they are color coded cluster signal distributions.

Author Response: Thank you for pointing this out. We clarified this in the new caption of Fig. 2:
> "Panels $A_2$ to $D_2$ show spatial distribution of pixels across FIB slices partitioned into four clusters based on pixel-specific spectral patterns at the Mn $L_{3,2}$ absorption edge."

[1.10]   Referee comment: (p 8, 31) 'While STXM-NEXAFS measurements are conducted with energies in the eV range, FIB preparation and SEM imaging utilize energies in the keV range.' The correlation of damage-potential and particle energy is an oversimplification. The X-rays transfer ALL their energy to the sample on absorption, whereas the ion and electron beams transfer only a portion. For ions it is momentum rather than energy transfer that is important I suspect. The reduction is probably done by liberated electrons.

Author Response: The paragraph referred to has been reworked to point out the very different energy transfer processes:

> "Generally, the varnish samples experienced an intense ion and electron bombardment as well as high X-ray exposure in the course of the preparation and analysis. Accordingly, all applied techniques – FIB, SEM, and STXM – are in principle potential sources for the beam damage (Süzer 2000, Bassim et al. 2012). The soft X-rays in STXM (~0.3 to 0.7 keV), accelerated electrons in SEM (~2 to 5 keV), and accelerated $Ga^+$ ions in FIB (~30 kV) are characterized by widely different energies. Moreover, their energies – and thus the potential damage – are deposited in the samples via different mechanistic pathways: Soft X-rays mostly act via core electron excitation up to an ionization of the atom, followed by a relaxation and filling of the core hole vacancy with associated photon and Auger electron emissions. As stated in the introduction, accelerated electrons mostly interact with varnish-like specimens via inelastic scattering, possibly causing radiolytic processes in the course of electronic excitations. Accelerated ions mostly act via nuclear, i.e. elastic collisions, resulting in sputtering, but electronic excitations should not be neglected. Our experiments showed, however, that the damaging effect of STXM is negligible: In dedicated tests, sequences of successive stack scans were recorded on the same area and no difference in the spectral patterns (i.e., at the absorption edges of Mn and other elements) could be observed. Moreover, previous X-ray microspectroscopy measurements have been successfully performed on materials with different Mn oxidation states (e.g., Bargar et al., 2001, Glasauer et al., 2006, Pecher et al., 2000, Pecher et al., 2003, Tebo et al., 2004, Toner et al., 2005)."

All changes requested and/or recommended by the referee in his minor comments have been implemented in the revised version of the manuscript.

References:

[revised manuscript text omitted]

---

## Author Comment (AC2) · 21 Jan 2019

Response to referee #2 (D. S. Macholdt et al., Artifacts from manganese reduction in rock samples prepared by focused ion beam (FIB) slicing for X-ray microspectroscopic analysis)

We appreciate the very thorough and helpful comments by referee #2, which have been considered carefully and helped to improve the quality of our manuscript. The referees' comments and our responses are outlined in detail below:

[1.1]    Referee comment: Page 8 line 32: "To verify whether a layer of modified material is actually distributed homogeneously on the surface of the sample" Why would you assume an even distribution? Please justify.

Author Response: In the course of our extensive rework of the manuscript, the above-mentioned statement was omitted. We must admit that we are not capable of proving how deep the damaged layer reaches or if the damaged layer thickness is similar across the sample. However, the spatial distribution is obviously even enough to show clearly the mentioned thickness dependency (refer to Fig. 2. and 3). We cannot exclude that areas with different chemical composition inside the varnish coating behave differently on ion beam exposure. Furthermore, the resolution of the here shown STXM stacks is not high enough to clearly resolve the submicron to nanometer-sized layering with good statistics, so any differences in beam damage sensitivity because of compositional fluctuations in the varnish got averaged and therefore remain invisible in our measurements.

[1.2]    Referee comment: Page 11 line 10:  "As there is at this time no alternative to FIB as sample preparation technique to produce intact ultra-thin slices of rock samples," The reader might wonder which are the benefits provided by FIB compared to, e.g., Argon ion slicing that has been also used in production of thin foils especially for TEM. According to this statement, you don't consider Argon ion slicing as an alternative to FIB. However, if this is the case especially for the samples used in this study, the reader would appreciate some reasoning.

Author Response: We appreciate this helpful comment by the referee. Indeed, Argon ion slicing could be a suitable alternative to prepare ultra-thin varnish slices. Accordingly, the section#

> "As there is at this time no alternative to FIB as sample preparation technique to produce intact ultra-thin slices of rock samples, one needs to be aware of these problems and choose preparation parameters that help to keep damage to a minimum. To reduce or minimize the damaged area, the preparation procedure could be conducted using lower voltages during preparation with the FIB and SEM or, if available, a cryo-FIB (Bassim et al. 2012). However, it is left to further studies to investigate whether oxidation states can indeed be kept unchanged using more gentle preparation approaches."

has been changed to

> "As FIB is a widely used technique to produce ultra-thin slices of rock samples, one needs to be aware of these problems and choose preparation parameters that help to keep damage to a minimum. To reduce or minimize the damaged volume, the preparation procedure could be conducted using not only low currents, but lower voltages during FIB preparation. In contrast, lowering the accelerating voltage in SEM might have an opposing, more damaging effect (Joy and Joy, 1996). If available, a cryo-FIB approach (Bassim et al. 2012) could be applied. Sezen et al. (2011) showed, however, that cryogenic conditions could not prevent or even slow down the degradation of conjugated polymers during FIB milling. Alternatively, Argon ion slicing (Stojic and Brenker, 2010) may be a more gentle and, therefore, suitable approach to reduce beam damage (e.g., Mn reduction) in the

preparation of ultrathin varnish slices. Even less damaging might be iodine ion milling as mentioned in Barber (1993). Fischione et al. (2017) established a method in which the damaged surface layers can be removed after FIB milling by a small spot Argon ion milling process. However, it is left to further studies to investigate whether oxidation states can indeed be kept unchanged using such more gentle preparation approaches."

[1.3]    Referee comment: Page 3 line 33: "Here we report about our findings observed during the investigation of the Mn oxidation states in 14 rock varnish samples, collected in different environments and countries." and Page 4 line 2: "For the sake of brevity, and since all samples showed the same phenomena, these findings will be exemplified using measurements on one of the samples." It is interesting that no differences between the varnishes were found especially as you have previously reported (Macholdt et al. 2017a) that layers of Mn-rich material and structures like cavities vary significantly between coatings of rock samples collected from different environments and regions. Perhaps you could refer to your earlier study to emphasize the importance of the finding of this manuscript - that the sample preparation of this sort produces similar kind of artifacts no matter what the structure of the varnish is.

Author Response: Thanks for this thought. As a response to comment [1.1] by referee 1, we included further plots from other varnish types into the manuscript text and discuss to what extent similar beam damage patterns have been observed for most samples, with few exceptions. Please refer to response to [1.1] for details.

[1.4]    Referee comment: Page 10 line 8: "we found that artifacts are produced during the preparation of the samples by FIB and monitoring by SEM, which create a high degree of uncertainty for oxidation state analyses." The reader would appreciate a quantitative estimate. Would it be possible to give a rough estimate on how much sample preparation of this kind adds to the total uncertainty – on the basis of the case presented in the manuscript?

Author Response: Unfortunately, our attempts to directly compare microtomes vs. FIB slides failed to provide a direct measure of beam damage (from FIB and/or SEM) relative to the native oxidation state. In lack of suitable reference substances and without a detailed understanding of the damaging mechanisms and their proportionate amounts of contribution, any quantitative statement would be highly speculative. Typical for ambient samples is their heterogeneity and diverse composition, which adds another dimension of uncertainty we do not oversee.

References:

Barber, D. J.: Radiation damage in ion-milled specimens: characteristics, effects and methods of damage limitation, Ultramicroscopy, 52(1), 101–125, doi:10.1016/0304-3991(93)90025-S, 1993.

Bassim, N. D., De Gregorio, B. T., Kilcoyne, A. L. D., Scott, K., Chou, T., Wirick, S., Cody, G. and Stroud, R. M.: Minimizing damage during FIB sample preparation of soft materials, J. Microsc., 245(3), 288–301, doi:10.1111/j.1365-2818.2011.03570.x, 2012.

Fischione, P. E., Williams, R. E. A., Genç, A., Fraser, H. L., Dunin-Borkowski, R. E., Luysberg, M., Bonifacio, C. S. and Kovács, A.: A Small Spot, Inert Gas, Ion Milling Process as a Complementary Technique to Focused Ion Beam Specimen Preparation, Microsc. Microanal., 23(4), 782–793, doi:10.1017/S1431927617000514, 2017.

Joy, D. C. and Joy, C. S.: Low voltage scanning electron microscopy, Micron, 27(3–4), 247–263, doi:10.1016/0968-4328(96)00023-6, 1996.

Macholdt, D. S., Jochum, K. P., Pöhlker, C., Arangio, A., Förster, J.-D., Stoll, B., Weis, U., Weber, B., Müller, M., Kappl, M., Shiraiwa, M., Kilcoyne, A. L. D., Weigand, M., Scholz, D., Haug, G. H., Al-Amri, A. and Andreae, M. O.: Characterization and differentiation of rock varnish types from different environments by microanalytical techniques, Chem. Geol., 459, 91–118, doi:10.1016/j.chemgeo.2017.04.009, 2017a.

Sezen, M., Plank, H., Fisslthaler, E., Chernev, B., Zankel, A., Tchernychova, E., Blümel, A., List, E. J. W., Groggera, W. and Pölta, P.: An investigation on focused electron/ion beam induced degradation mechanisms of conjugated polymers, Phys. Chem. Chem. Phys., (13), 20235–20240, doi:10.1039/C1CP22406A, 2011.

Stojic, A. N. and Brenker, F. E.: Argon ion slicing (ArIS): a new tool to prepare super large TEM thin films from Earth and planetary materials, Eur. J. Mineral., 22(1), 17–21, doi:10.1127/0935-1221/2009/0022-2004, 2010.